# HCDA: Hierarchical Contrastive Learning with Dynamic Attention Fusion

## Abstract

Multi-view clustering integrates complementary features and semantics across modalities to provide comprehensive data representations. However, existing methods struggle with semantic fragmentation and inefficient view integration when processing complex multi-view data. To address these challenges, this paper proposes a Hierarchical Contrastive learning and Dynamic Attention (HCDA) fusion framework. HCDA achieves fine-grained semantic decoupling by constructing multi-level semantic spaces (independent semantic space, cross-view fused semantic space, and global semantic space), and designs a dual-path attention mechanism to dynamically allocate view weights based on global semantics, suppressing interference from low-quality views while enhancing the dominant role of high-contribution views. Additionally, HCDA jointly optimizes contrastive losses between cross-view fused features and independent features to balance view commonality and specificity, enhancing both discriminability and consistency. Experimental results demonstrate HCDA's superiority over state-of-the-art methods on benchmark datasets. Ablation studies confirm the efficacy of hierarchical semantic modeling and dynamic weighting, while parameter analysis highlights robustness to hyperparameter variations.

## 1 Introduction

Multi-view data exhibits complementary features and semantics across modalities, providing comprehensive data representations through cross-source collaboration. Deep Multi-View Clustering (DMVC) leverages these complementary and consistent relationships to integrate multi-source data and achieve unsupervised semantic discovery Chao et al. (2021). Contrastive learning balances both complementarity and consistency between views by aligning their representations and decoupling shared and view-specific features. The recent integration of contrastive learning with DMVC has significantly improved clustering performance on complex multi-view data, through self-supervised signal generation and cross-view relation modeling.

DMVC methods based on contrastive learning can be categorized into four types: vanilla contrastive Multi-view Clustering (MvC) methods Hassani & Khasahmadi (2020); Chen et al. (2020), robust MvC for incomplete instances Lin et al. (2021); Wang et al. (2018); Li et al. (2023), robust MvC against false negatives Cui et al. (2024); Guo et al. (2024), and multi-level contrastive methods Chen et al. (2025); Zhang & Che (2024); Wang et al. (2023). Most contrastive learning techniques aim to map multi-view data into a shared latent semantic space, thereby capturing cross-view consistency and complementarity. This approach directly models global semantic consistency between views, which is most effective when the data exhibits strong inter-view correlations Tian et al. (2020); Hu et al. (2023). In contrast, some methods focus on preserving view-specific semantics to manage heterogeneity, offering robustness against noisy or low-quality views by maintaining discriminative features unique to each view Ren et al. (2024). State-of-the-art methods often combine these two strategies to achieve fine-grained semantic fusion Yan et al. (2023); Chen et al. (2025). By balancing shared semantics and view-specific semantics, contrastive learning methods can improve clustering performance. In multi-view learning, significant differences in semantic relevance among views, such as the contrast between strongly correlated and weakly correlated perspectives, can undermine the effective integration of a unified semantic space. Additionally, overemphasizing shared semantics across views risks oversimplifying unique details inherent to individual perspectives, while relying excessively on view-specific features may introduce redundant model parameters. This dual

challenge highlights the critical need to balance common semantics and view-specific information in multi-view representation learning.

Based on the above analysis, current deep multi-view contrastive learning methods face two key challenges. The first challenge is semantic fragmentation, caused by independent view modeling, where the neglect of complementary low-level semantic relationships across views leads to inconsistent feature alignment and limits representation discriminability, particularly due to insufficient modeling of fine-grained semantic interactions and hierarchical alignment. The second challenge is ineffective multi-view integration, where traditional approaches' independent feature processing results in redundant feature spaces and semantic misalignment, exacerbated by naive statistical weighting strategies that fail to distinguish critical semantics from noise, the lack of global consistency constraints, and isolated weight allocation mechanisms that cannot capture cross-view semantic dependencies. These limitations collectively hinder optimal representation learning and downstream task performance.

To address the aforementioned challenges, this paper proposes a Hierarchical Contrastive learning framework with Dynamic Attention (HCDA) fusion for multi-view clustering. First, multi-view data is encoded into latent features, and a multi-granularity semantic space is constructed through cross-view interactive fusion, global feature concatenation, and independent view mapping. Next, a dual-path attention mechanism is designed for semantics-driven dynamic weight allocation. Using global semantics as a reference, it adaptively learns weights for cross-view fused features and independent features, suppressing interference from low-quality views while enhancing the dominant role of high-contribution views. Finally, the contrastive losses between cross-view fused features and global semantics and between independent features and global semantics are jointly optimized. Through iterative learning, a balance between commonality and specificity is achieved. By employing hierarchical semantic decoupling and adaptive contrastive objectives, this method breaks through the reliance of traditional approaches on static fusion and single-objective optimization, significantly improving clustering robustness in complex multi-view scenarios.

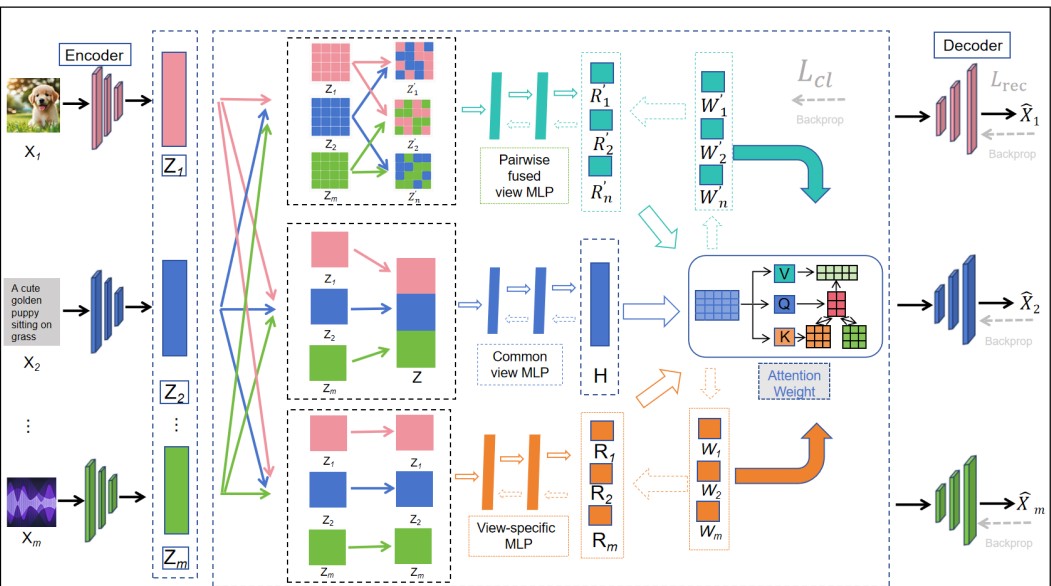

Figure 1: HCDA encodes multi-view data, builds a semantic space via fusion and mapping, and uses dual-path attention to weight views. Contrastive losses balance shared and unique features for robust clustering.

The main contributions of this paper are summarized as follows:

- By generating multi-granularity features through multi-layer heterogeneous semantic spaces, the method explicitly captures local complementarity, global consistency, and view-specific characteristics.

- A dual-path attention mechanism is introduced to dynamically learn weights for cross-fused features and independent features, adaptively suppressing interference from redundant views while enhancing the semantic dominance of high-contribution views.

- A cross-hierarchical contrastive learning framework is designed to jointly optimize contrastive losses between cross-fused features and global semantics, as well as between independent view features and global semantics, thereby reinforcing local collaboration and global consistency.

- Extensive experimentation across real benchmark datasets serves to highlight both the superiority and efficiency of the proposed HCDA method.

## 2 RELATED WORK

The inherent pairing characteristic of multi-view data renders the contrastive learning paradigm a natural fit for MvC. Existing MvC methods can be roughly classified into the following four categories: i) Deep embedding-based MVC Xu et al. (2021a; 2024); Xia et al. (2022), which utilizes autoencoders (AEs) or variational autoencoders (VAEs) to learn low-dimensional embeddings while directly optimizing clustering objectives such as KL divergence. ii) Deep graph-based MVC Du et al. (2023); Rahman et al. (2023); Ke et al. (2024); Wang et al. (2024), which leverages graph neural networks (GNNs) to model structural relationships across views. iii) Adversarial-based MVC Zhou & Shen (2020); Junpeng et al.; Huang et al. (2024); Yang et al. (2024), which employs generative adversarial networks (GANs) or adversarial training to align cross-view distributions. iv) Contrastive learning-based MVC Jiang et al. (2024); Zheng et al. (2024), which enhances discriminative clustering by maximizing agreement between positive pairs (same instance, different views) and minimizing agreement for negative pairs.

In addition to these broad categories, contrastive multi-view clustering methods can be further categorized into the following approaches: i) Vanilla contrastive MvC methods Hassani & Khasahmadi (2020); Chen et al. (2020), which directly exploit contrastive learning to enhance the discrimination of learned representations by maximizing the mutual information between distinct views. ii) Robust contrastive MvC for incomplete instances Lin et al. (2021); Wang et al. (2018); Li et al. (2023), which ensures robustness by reconstructing missing views or adapting contrastive objectives to incomplete data. iii) Robust contrastive MvC against false negatives Cui et al. (2024); Guo et al. (2024), which redesigns dedicated loss functions or similarity estimation techniques to address false negatives inherent in contrastive learning, thus improving clustering performance. iv) Multi-level contrastive methods Chen et al. (2025); Zhang & Che (2024); Wang et al. (2023), which perform contrastive learning at multiple granularities (e.g., instance-level, cluster-level, or feature-level) to capture hierarchical consistency. These approaches collectively enhance view consistency, semantic alignment, and negative-pair reliability, driving advancements in multi-view clustering performance across diverse scenarios.

## 3 METHOD

In this section, we present HCDA, a hierarchical contrastive learning framework with dynamic attention fusion for multi-view clustering. Given multi-view inputs $\{\mathbf{X}_1, \ldots, \mathbf{X}_M\}$, each view is encoded into a latent representation $\{\mathbf{Z}_1, \ldots, \mathbf{Z}_M\}$ through view-specific encoders. Based on these, HCDA constructs a multi-granularity semantic space: the independent semantics $\mathbf{R}_v$ are obtained by feeding $\mathbf{Z}_v$ into view-specific MLPs; the fused semantics $\mathbf{R}'_v$ are generated by pairwise fusion of $\mathbf{Z}_v$ followed by a shared MLP; and the global semantics $\mathbf{H}$ are learned by concatenating all $\mathbf{Z}_v$ and projecting them via a common MLP. To dynamically assess the contribution of each view, a semantics-driven dual-path attention mechanism computes attention weights by referencing the global semantic $\mathbf{H}$, where the weights for independent and fused semantics are defined as $W^{\text{indep}} = \text{Attn}(\mathbf{R}_v, \mathbf{H})$ and $W^{\text{fuse}} = \text{Attn}(\mathbf{R}'_v, \mathbf{H})$, respectively. These weights modulate the contrastive learning process by enhancing informative views while suppressing noisy ones. The overall training objective integrates two losses: the reconstruction loss $\mathcal{L}_{\text{rec}}$ for data fidelity, and the contrastive loss $\mathcal{L}_{\text{cl}}$ that jointly optimizes the alignment between $\mathbf{R}_v$, $\mathbf{R}'_v$ and the global representation $\mathbf{H}$, forming the total loss:

$$\mathcal{L}_{\text{total}} = \mathcal{L}_{\text{rec}} + \mathcal{L}_{\text{cl}}. \tag{1}$$

Trained end-to-end, HCDA progressively enhances both view-level specificity and global consistency, leading to robust and discriminative clustering representations across diverse scenarios.

### 3.1 HIERARCHICAL SEMANTIC SPACE CONSTRUCTION

To effectively disentangle shared and view-specific information in multi-view scenarios, we propose a hierarchical semantic representation framework that comprises three levels: view-specific semantics, cross-view fused semantics, and global semantics. This multi-level representation enables the model to systematically capture both the inherent characteristics of individual views and their complementary information across views, thereby enhancing the capacity for multi-view learning.

Given the latent representations $\{\mathbf{Z}_v\}_{v=1}^M$ extracted by view-specific encoders, where each $\mathbf{Z}_v \in \mathbb{R}^d$ denotes the low-level representation of the $v$-th view, we first model view-consistent semantics via dedicated projection networks. Specifically, each latent representation $\mathbf{Z}_v$ is fed into a view-specific projection network to obtain the semantic representation $\mathbf{R}_v = \text{MLP}_v(\mathbf{Z}_v; \Psi_v)$, which captures the consensus information inherent in the $v$-th view. These individual view-specific semantic representations help capture the distinctive features of each view, enabling the model to better understand the nuances within each modality.

In order to further exploit the complementarity between different views, we perform an L2 norm-guided weighted fusion on each pair of latent features $\mathbf{Z}_a$ and $\mathbf{Z}_b$ ($a \neq b$) before semantic modeling. This fusion strategy ensures that views with stronger features (in terms of their L2 norm) contribute more significantly to the fused representation, promoting a more balanced integration of multi-view information. The fused intermediate representation $\mathbf{Z}_{ab}$ is computed as:

$$\mathbf{Z}_{ab} = \frac{\|\mathbf{Z}_a\|}{\|\mathbf{Z}_a\| + \|\mathbf{Z}_b\|}\mathbf{Z}_a + \frac{\|\mathbf{Z}_b\|}{\|\mathbf{Z}_a\| + \|\mathbf{Z}_b\|}\mathbf{Z}_b. \tag{2}$$

This L2 norm-based fusion approach ensures that views with more prominent features play a greater role in the combined representation, while still preserving the contribution of other views. The fused representation $\mathbf{Z}_{ab}$ is then passed through a shared Multi-Layer Perceptron (MLP) to extract cross-view fused semantics $\mathbf{R}'_{ab} = \text{MLP}(\mathbf{Z}_{ab}; \Psi)$, which encodes the complementary features between views. This cross-view fusion helps the model learn how the different views interact and contribute to a more holistic understanding of the data.

Moreover, to obtain a unified global semantic anchor that encapsulates the entire multi-view information, we concatenate all latent features $[\mathbf{Z}_1; \mathbf{Z}_2; \ldots; \mathbf{Z}_M]$ and pass them through a global projection network to generate the global semantic representation $\mathbf{H} = \text{MLP}_{\text{common}}([\mathbf{Z}_1; \mathbf{Z}_2; \ldots; \mathbf{Z}_M]; \Psi_g)$, which aligns semantics across different views. This global representation serves as the anchor for guiding the model's learning process, providing a unified semantic space in which all views are aligned and comparable. The global semantic anchor $\mathbf{H}$ helps the model establish a common reference point for all views, enabling consistent learning and clustering.

By jointly constructing view-specific, cross-view fused, and global semantic spaces, our model hierarchically abstracts multi-view features from intra-view consistency to inter-view complementarity and global alignment. This structure not only enhances view discrimination but also ensures effective fusion of local and global contexts, providing a strong basis for attention-guided fusion and contrastive learning. The hierarchical design promotes rich semantic representations that integrate fine-grained view details with shared structural patterns.

### 3.2 DUAL-PATH ATTENTION-WEIGHTED CONTRASTIVE LEARNING

In order to effectively capture both individual semantics from each view and the cross-view complementarity, we propose a dual-path contrastive learning strategy, enhanced by attention-guided weighting. This strategy aims to balance the individual characteristics of each view with the global semantic representation, improving the ability of the model to distinguish between distinct data points and fuse complementary information across multiple views.

Specifically, we first assign a contrastive importance score to each view using an attention mechanism. These importance scores are then used to weight the contrastive loss for each view, ensuring

that views with higher relevance to the global semantics receive more emphasis during optimization. The overall framework is designed to optimize two levels of contrastive learning objectives: single-view semantic contrast and cross-view fusion contrast. This two-level learning approach ensures that the model simultaneously learns the semantics of individual views and their relationships with the global representation, thereby improving the robustness and accuracy of the multi-view clustering task.

**Hierarchical Cross-View Attention Weighting**: The core of effectively capturing cross-view relationships lies in the design of the attention mechanism. Let $\mathbf{H} \in \mathbb{R}^{N \times d}$ be the global semantic representation matrix, where $N$ represents the number of samples and $d$ is the feature dimension. To begin, we project $\mathbf{H}$ into a query vector $\mathbf{Q} \in \mathbb{R}^{d_v}$ via a learnable projection matrix $\mathbf{W}_q \in \mathbb{R}^{d \times d_v}$, defined as $\mathbf{Q} = \mathbf{H}\mathbf{W}_q$, where $d_v$ is the dimension of the view-specific representation.

For each view $v$, we have the view-specific representation $\mathbf{R}_v \in \mathbb{R}^{N \times d}$, which is projected into a view-specific key vector $\mathbf{K}_v \in \mathbb{R}^{N \times d_v}$ using another learnable matrix $\mathbf{W}_k \in \mathbb{R}^{d \times d_v}$, i.e., $\mathbf{K}_v = \mathbf{R}_v\mathbf{W}_k$. Next, we compute the attention score matrix $\mathbf{A}_v$ based on the scaled dot-product attention, which measures the alignment between the global semantic query $\mathbf{Q}$ and the view-specific key $\mathbf{K}_v$. Specifically, the attention score matrix $\mathbf{A}_v \in \mathbb{R}^{N \times N}$ is computed as the scaled dot-product of the view-specific key matrix $\mathbf{K}_v$ and the transpose of the global query matrix $\mathbf{Q}$, normalized by the square root of the dimension $d_v$ of the view-specific key, i.e., the attention score between the $i$-th data point's view-specific key $\mathbf{K}_v^{(i)}$ and the global query $\mathbf{Q}^{(j)}$ is $\mathbf{A}_v^{(i,j)} = \mathbf{K}_v^{(i)}\mathbf{Q}^{(j)T}/\sqrt{d_v}$.

To capture the global relationship between all pairs of samples and normalize the attention scores across all views, we define the normalized attention weight $W_v$ for view $v$ as:

$$W_v = \frac{\exp\left(\sum_{i=1}^{N} \exp\left(\frac{1}{\sqrt{d_v}}\sum_{j=1}^{N}\mathbf{A}_v^{(i,j)}\right)\right)}{\sum_{v=1}^{M}\exp\left(\sum_{i=1}^{N}\exp\left(\frac{1}{\sqrt{d_v}}\sum_{j=1}^{N}\mathbf{A}_v^{(i,j)}\right)\right)}. \tag{3}$$

where $\mathbf{A}_v^{(i,j)}$ represents the attention score between the $i$-th data point's view-specific key and the global query. The inner exponential sum captures the influence of each individual element $\mathbf{A}_v^{(i,j)}$, and the outer exponential operation enhances the importance of larger values, reflecting stronger alignment with the global semantic space. The normalization ensures that the total attention weights across all views sum to 1, making the contributions of each view comparable.

This attention mechanism allows each view's contribution to be weighted according to its relevance to the global semantic space. Higher attention scores indicate stronger alignment with the global representation, and thus, greater influence on the model during training.

**Contrastive Loss Design**: After obtaining the attention weights for each view, we design a unified contrastive loss that simultaneously handles both single-view and fusion-view contrastive losses. The goal of this contrastive loss is to maximize the agreement between the representations of the same data point across different views, while simultaneously minimizing the similarity between representations of different data points. For each view $v$, we consider the semantic embedding $\mathbf{R}_v^{(i)}$ of a given sample and the global anchor $\mathbf{H}^{(i)}$ as a positive pair, while all other embeddings from different samples in the dataset form negative pairs. The similarity between the view embedding $\mathbf{R}_{v_j}$ and the global anchor $\mathbf{H}_i$ is defined using cosine similarity as:

$$\text{sim}(\mathbf{H}_i, \mathbf{R}_{v_j}) = \frac{\langle \mathbf{H}_i, \mathbf{R}_{v_j}\rangle}{\|\mathbf{H}_i\| \cdot \|\mathbf{R}_{v_j}\|}. \tag{4}$$

This cosine similarity measures how similar two vectors are in the high-dimensional feature space, where $\mathbf{H}_i$ denotes the global semantic representation and $\mathbf{R}_{v_j}$ denotes the view-specific embedding. To simultaneously optimize both the single-view representations and the fusion-view representations, we propose a unified contrastive loss, defined as:

$$\mathcal{L}_{\text{cl}}^{(v)} = -\frac{1}{N}\sum_{i=1}^{N}\left[\alpha \log \frac{e^{\text{sim}(\mathbf{R}_v^{(i)}, \mathbf{H}^{(i)})/\tau}}{\sum_{j=1, j\neq i}^{N} e^{\text{sim}(\mathbf{R}_v^{(i)}, \mathbf{H}^{(j)})/\tau}} + (1-\alpha)\log\frac{e^{\text{sim}(\mathbf{R'}_v^{(i)}, \mathbf{H}^{(i)})/\tau}}{\sum_{j=1, j\neq i}^{N} e^{\text{sim}(\mathbf{R'}_v^{(i)}, \mathbf{H}^{(j)})/\tau}}\right]. \tag{5}$$

Here, $\alpha \in [0,1]$ is a custom hyperparameter that balances the contributions of the single-view and fusion-view contrastive terms. This joint loss formulation encourages both the view-specific and fused representations to align closely with the global semantic anchor, thus capturing the complementary semantics across multiple views while preserving the discriminative structure necessary for downstream clustering tasks.

**Final Objective Function**: The overall objective function is designed to balance the reconstruction loss $\mathcal{L}_{rec}$, the attention-weighted single-view contrastive loss, and the fusion contrastive loss. By integrating these losses, we ensure that the model learns to capture both the individual semantics of each view and the global semantic structure while preserving the relationships across different views.

The total objective function is expressed as:

$$\mathcal{L}_{total} = \sum_{v=1}^{M} \mathcal{L}_{rec}^{(v)} + \sum_{v=1}^{M} W_v \cdot \mathcal{L}_{cl}^{(v)}, \tag{6}$$

In this formulation, the reconstruction loss $\mathcal{L}_{rec}$ ensures preservation of essential features from each view by minimizing the difference between the original input $\mathbf{X}_v$ and its reconstruction $\hat{\mathbf{X}}_v$. It is formally defined as:

$$\mathcal{L}_{rec} = \sum_{v=1}^{V} \left\| \mathbf{X}_v - \hat{\mathbf{X}}_v \right\|_F^2, \tag{7}$$

where $V$ denotes the number of views and $\| \cdot \|_F$ is the Frobenius norm.

The contrastive loss enforces alignment between the global representation and view-specific representations, promoting semantic consistency across views. Attention weights $\alpha_v$ dynamically adjust the importance of each view based on its relevance to the global semantic space, allowing the model to focus on more informative views for clustering.

This dynamic weighting mechanism improves the model's ability to leverage complementary information from multiple views, thereby enhancing the robustness and accuracy of multi-view clustering.

The training process of HCDA involves pre-training for feature extraction, feature fusion, cross-view attention, contrastive learning, and clustering, as detailed in the following algorithm.

---

**Algorithm 1** HCDA Model Training Process

---

**Require:** Multi-view dataset $\{\mathbf{X}_v\}_{v=1}^{V}$, number of epochs $T$
**Ensure:** Clustering indicator matrix $\mathbf{U}$
1: **for** $t = 1$ to $T$ **do**
2:     Train autoencoders to obtain latent features $\{\mathbf{Z}_v\}_{v=1}^{V}$
3:     Perform pairwise L2-norm fusion on $\{\mathbf{Z}_v\}$ to get $\mathbf{Z}'$ (Eq. 2)
4:     Compute attention scores $W_v$ from $\mathbf{R}_v$, $\mathbf{R}'$, and $\mathbf{H}$ using the attention mechanism (Eq. 3)
5:     Compute unified contrastive loss $\mathcal{L}_{contrast}$ (Eq. 5)
6:     Update model parameters by optimizing total loss $\mathcal{L}_{total}$
7: **end for**
8: Perform clustering on global representation $\mathbf{H}$ to obtain $\mathbf{U}$

---

# 4 EXPERIMENTS

## 4.1 EXPERIMENTAL SETUP

**Dataset Description:** We conduct experiments on five publicly available multi-view datasets across diverse domains to evaluate the generalization capability and robustness of our method: **Fashion** Xu et al. (2021b) is a dataset of fashion product images with multiple views, commonly used for image classification tasks. **Synthetic3D** Kumar et al. (2011) is a collection of synthetic 3D object models rendered from different viewpoints, used to evaluate view-invariant representation learning.

**100leaves** Li et al. (2021) is a leaf classification dataset containing images from 100 plant species, providing multiple views per instance. **Caltech-5V** Fei-Fei et al. (2004) is a benchmark dataset for multi-view object recognition, containing five visual features extracted from the Caltech101 dataset. **CIFAR-10** Chauhan et al. (2018) is a widely used 10-class image classification dataset, where different feature extractors generate multiple views.

**Comparison Methods:** We compare our method (HCDA) with ten recent deep multi-view clustering approaches. **CSOT** Zhang et al. (2024) aligns cross-view semantics via optimal transport with semantic-aware reweighting. **DealMVC** Yang et al. (2023) integrates cross-view features and pseudo-label graphs through global–local discriminative learning. **SCM** Luo et al. (2024) applies contrastive learning with graph-based metric alignment. **DDMVC** Xu et al. (2025) enhances clustering by discriminative latent learning. **MFLVC** Xu et al. (2022) introduces an effective fusion strategy for feature and view learning. **CVCL** Chen et al. (2023) designs large-scale contrastive learning for multi-view clustering. **MAGA** Bian et al. (2024) combines multi-modal learning with attention mechanisms. **SCMVC** Wu et al. (2024) integrates spectral clustering into a contrastive framework. **DSMVC** Tang & Liu (2022) improves embeddings via self-supervised multi-view learning. **SEM** Xu et al. (2023) embeds semantic representations into contrastive learning to boost performance.

**Implementation Details:** All datasets are reshaped into vectors, and fully connected (Fc) autoencoders with a similar architecture are used to extract low-level features $\{\mathbf{Z}_v\}_{v=1}^M$. Specifically, for each view, the structure of the encoder is: Input - Fc500 - Fc500 - Fc2000 - Fc64, and the decoder is symmetric with the encoder. After that, a linear MLP with the structure Input(64) - Fc20 is used to extract view-consensus features $\{\mathbf{R}_v\}_{v=1}^M$, and another non-linear MLP with a two-layer architecture, i.e., Input($M \times 64$) - Fc256 - Fc20, is used to learn global feature representations $\mathbf{H}$. After feature fusion, the fused $\mathbf{Z}$ is passed through an MLP, with a structure from 64 to 20 dimensions, to learn multi-level semantic space. The following settings are consistent across all experimental datasets. The ReLU activation function is used in all layers except the output layer. Adam is chosen as the optimizer with a default learning rate of 0.0003. The experiments are conducted on a Windows PC with an Intel(R) Core(TM) i7-12700K CPU @ 3.6 GHz, 32.0 GB RAM, PyTorch version 1.10.2, and an NVIDIA GeForce RTX 4070 GPU with 12 GB VRAM.

## 4.2 COMPARATIVE EXPERIMENTAL RESULTS ANALYSIS

The comparative experimental results on six multi-view datasets (Table 1) demonstrate that our proposed HCDA method achieves superior or highly competitive performance across all metrics (ACC, NMI, PUR). In particular, HCDA ranks first on five datasets such as **Fashion** and **100leaves**, and obtains the second-best NMI on **Synthetic3d**. This strong performance benefits from the proposed hierarchical semantic modeling framework, which captures local complementarity, global consistency, and view specificity through three heterogeneous feature spaces. The dual-path attention mechanism dynamically balances the contributions of fused and independent features, suppressing noisy views and enhancing discriminative cues. For instance, HCDA achieves 99.37% ACC and 98.35% NMI on **Fashion**, outperforming other methods by 2–5 percentage points. Moreover, the dual contrastive learning strategy jointly aligns cross-view and independent features with global semantics, effectively enhancing robustness and semantic disentanglement. On **100leaves**, HCDA surpasses existing methods like DealMVC and DSMVC with an ACC improvement of 9–10 points. These results validate the effectiveness of our design in handling diverse view semantics and low-quality views.

## 4.3 MODEL ANALYSIS

**Clustering Results Visualization Analysis**: To provide an intuitive demonstration of the clustering effect of the global features extracted by our method, we performed 2D visualization using t-SNE on the global features $\mathbf{H}$ from the Fashion dataset. The results show that as the training epochs increase, inter-class separability and intra-class compactness progressively improve, particularly at the 10th and 20th epochs, where the clustering effect is significantly optimized. This improvement can be attributed to our method's hierarchical high-order semantic space construction and dynamic fusion strategy, which captures local complementarity, global consistency, and view specificity through three heterogeneous semantic spaces. Specifically, the hierarchical semantic space construction allows the features of each view to be processed at multiple levels, while the dynamic fusion method

Table 1: Comparison of clustering performance (ACC, NMI, PUR) of different methods on five datasets.

| Method | Fashion | | | Synthetic3d | | | 100leaves | | | Caltech-5V | | | Cifar10 | | |
|---|---|---|---|---|---|---|---|---|---|---|---|---|---|---|---|
| | ACC | NMI | PUR | ACC | NMI | PUR | ACC | NMI | PUR | ACC | NMI | PUR | ACC | NMI | PUR |
| CSOT (2024) | 99.14 | 97.86 | 99.14 | 96.83 | 86.60 | 96.83 | 26.19 | 58.16 | 28.31 | 87.21 | 79.13 | 87.21 | 99.18 | 97.73 | 99.18 |
| DealMVC (2023) | 82.07 | 91.44 | 82.63 | 80.67 | 56.54 | 80.67 | 8.00 | 41.13 | 8.06 | 86.86 | 78.89 | 86.86 | 96.91 | 97.12 | 96.91 |
| SCM (2024) | 81.86 | 78.16 | 82.03 | 94.50 | 79.86 | 94.50 | 33.25 | 65.47 | 36.06 | 74.79 | 65.67 | 74.79 | 99.27 | 97.99 | 99.27 |
| DDMVC (2025) | 77.17 | 77.59 | 77.48 | 77.83 | 46.27 | 77.83 | 25.37 | 60.35 | 26.12 | 73.79 | 66.39 | 73.79 | 98.62 | 96.35 | 98.62 |
| MFLVC (2022) | 99.16 | 97.95 | 99.16 | 95.67 | 83.67 | 95.67 | 24.75 | 60.81 | 26.12 | 85.36 | 75.58 | 85.36 | 99.11 | 97.72 | 99.11 |
| CVCL (2023) | 99.31 | 98.20 | 99.31 | 89.20 | 67.17 | 89.20 | 18.81 | 51.57 | 19.56 | 84.29 | 72.36 | 84.29 | 15.50 | 48.96 | 16.00 |
| MAGA (2024) | 98.85 | 97.26 | 98.85 | 95.50 | 82.16 | 95.50 | 43.50 | 71.61 | 46.12 | 70.14 | 66.51 | 73.29 | 98.44 | 95.74 | 98.44 |
| SCMVC (2024) | 99.34 | 98.28 | 99.34 | 97.00 | **87.04** | 97.00 | 59.13 | 79.99 | 62.00 | 89.29 | 81.92 | 89.29 | 99.51 | 98.58 | 99.51 |
| DSMVC (2022) | 75.81 | 71.96 | 75.81 | 77.33 | 45.41 | 77.33 | 52.81 | 80.44 | 53.81 | 64.93 | 55.52 | 69.14 | 72.13 | 64.96 | 72.13 |
| SEM (2023) | 99.29 | 98.19 | 99.29 | 95.83 | 82.91 | 95.83 | 61.50 | 81.57 | 65.00 | 87.14 | 79.25 | 87.14 | 99.23 | 97.89 | 99.23 |
| HCDA | **99.37** | **98.35** | **99.37** | **97.00** | 87.01 | **97.00** | **71.56** | **85.78** | **73.44** | **91.50** | **84.31** | **91.50** | **99.53** | **98.66** | **99.53** |

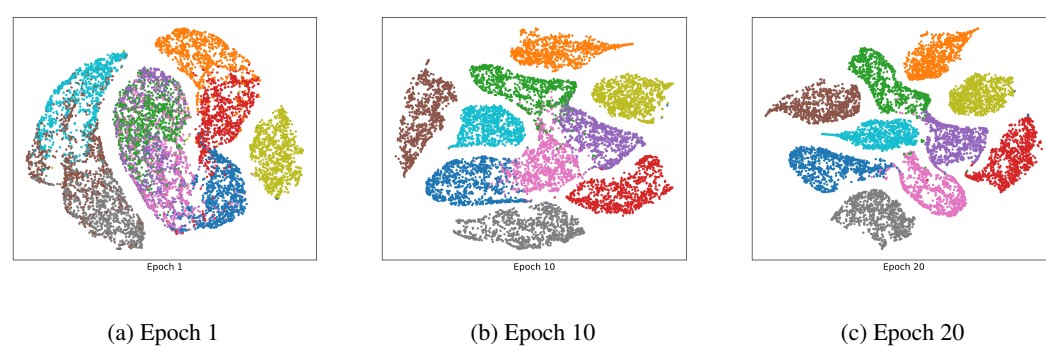

(a) Epoch 1        (b) Epoch 10        (c) Epoch 20

Figure 2: Clustering result visualization on the Fashion dataset at different training stages.

enables the model to fuse information across different levels, thereby enhancing the complementarity between views.

**Convergence Analysis**: Figure 3 shows the change trend of the loss value and clustering metrics during training. It can be observed that the loss function value gradually decreases as training progresses and eventually converges. Clustering performance metrics (such as ACC and NMI) rise rapidly in the early stages and stabilize later, further verifying the convergence and stability of our model.

**Hyperparameter Sensitivity Analysis**: We conduct a sensitivity analysis on the temperature coefficient $\tau$ in the global contrastive loss, varying $\tau \in \{0.1, 0.3, 0.5, 0.7, 1.0\}$ and evaluating clustering performance using ACC, NMI, and PUR (Figure 4). Results show that a small $\tau$ (e.g., 0.1) degrades performance by limiting semantic separability, while larger values (0.5–1.0) yield more stable and improved results, indicating that an appropriate $\tau$ is critical for balancing global and local information.

### 4.4 ABLATION EXPERIMENTS

The ablation study (Table 2) demonstrates that the feature fusion module (M2) significantly improves clustering performance compared to (M1), with notable increases in ACC on the Caltech-5V dataset, where ACC improves from 80.14 to 88.00, NMI from 68.84 to 78.89, and PUR from 80.14 to 88.00. Adding the attention weighting mechanism (M3) further enhances performance, with ACC on the 100leaves dataset increasing from 58.94 to 69.19, on Caltech-5V reaching 89.43, and on the Synthetic3d dataset reaching 96.83. Finally, our complete method (HCDA) achieves the best results across all datasets, with an ACC of 91.50 on Caltech-5V, 71.56 on 100leaves, and 97.00 on Synthetic3d, demonstrating the effectiveness of combining reconstruction, feature fusion, and attention weighting in improving clustering quality.

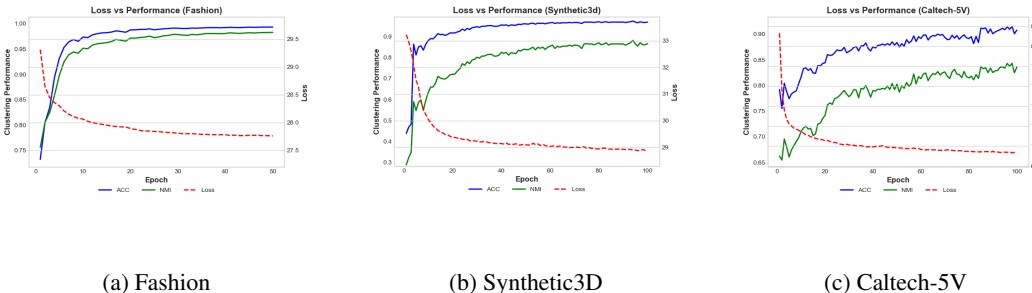

(a) Fashion                    (b) Synthetic3D                    (c) Caltech-5V

Figure 3: Evaluation metrics as a function of training epochs.

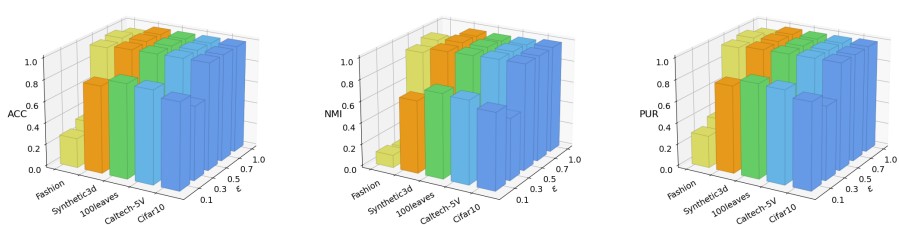

Figure 4: Impact of different temperature coefficient $\tau$ on clustering performance (ACC, NMI, PUR)

## 5 CONCLUSION

In this paper, we propose **HCDA**, a Hierarchical Contrastive Learning framework with Dynamic Attention Fusion for multi-view clustering. HCDA tackles semantic fragmentation and suboptimal view integration by constructing a multi-granularity semantic space and employing a dual-path attention mechanism for adaptive view weighting. Through hierarchical fusion—comprising cross-view interaction, global aggregation, and independent mapping—HCDA effectively balances shared and view-specific semantics. The contrastive learning scheme jointly optimizes fused and independent representations with respect to global semantics, enhancing feature discrimination and consistency. Compared to prior methods, HCDA offers a more structured and flexible fusion strategy, leading to improved scalability and robustness. Extensive experiments demonstrate that HCDA consistently outperforms state-of-the-art approaches in clustering accuracy, robustness, and stability.

**Limitation and Future Work.** While effective, HCDA's pairwise fusion design may increase computational cost as the number of views grows. Future work includes extending HCDA to dynamic or streaming settings and incorporating advanced semantic alignment techniques to further enhance generalization.

Table 2: Ablation Study Results on 100leaves, Caltech-5V, and Synthetic3d Datasets.

| Modules | $L_{rec}$ | $L_{cl}$ | $W$ | 100leaves | | | Caltech-5V | | | Synthetic3d | | |
|---------|-----------|----------|-----|------|------|------|------|------|------|------|------|------|
| | | | | ACC | NMI | PUR | ACC | NMI | PUR | ACC | NMI | PUR |
| M1 | ✓ | | | 58.94 | 79.74 | 63.19 | 80.14 | 68.84 | 80.14 | 68.50 | 33.67 | 68.50 |
| M2 | ✓ | ✓ | | 60.50 | 80.57 | 65.27 | 88.00 | 78.89 | 88.00 | 95.17 | 81.39 | 95.17 |
| M3 | ✓ | | ✓ | 69.19 | 86.52 | 72.56 | 89.43 | 82.29 | 89.43 | 96.83 | 86.51 | 96.83 |
| HCDA | ✓ | ✓ | ✓ | 71.56 | 85.78 | 73.44 | 91.50 | 84.31 | 91.50 | 97.00 | 87.01 | 97.00 |

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
