# OpenReview forum: "HCDA: Hierarchical Contrastive Learning with Dynamic Attention Fusion"
_ICLR.cc/2026/Conference — ICLR 2026 Conference Withdrawn Submission_

### Official Review · Reviewer_XYaH · 2025-10-24

**Soundness:** 2
**Presentation:** 3
**Contribution:** 2
**Rating:** 4
**Confidence:** 4

**Summary:**

This paper proposes a multi-view clustering framework named HCDA, which aims to address the issues of semantic fragmentation and inefficient view integration faced by existing methods when processing complex multi-view data. The authors achieve fine-grained semantic decoupling by constructing a multi-level semantic space. On this basis, a dual-path attention mechanism is designed, which utilizes global semantics as a reference to dynamically allocate weights to different views, thereby suppressing interference from low-quality views and enhancing the role of high-contribution views. The experimental section validates the superiority of the method on multiple benchmark datasets and demonstrates the effectiveness and robustness of each module through ablation studies and parameter analysis.

**Strengths:**

The core strength of the HCDA method lies in the organic integration of its hierarchical semantic modeling and dynamic attention fusion mechanism. Firstly, by explicitly constructing a three-level semantic space, the model can systematically capture intra-view consistency, inter-view complementarity, and global alignment, avoiding the semantic oversimplification or information loss caused by a single semantic space in traditional methods. Secondly, the proposed dual-path attention mechanism possesses strong semantic-driven capabilities; its weight calculation relies on the alignment degree between the global semantic $\mathbf{H}$ and each view representation, rather than simple statistical features, thus achieving more discriminative view weighting. This mechanism effectively solves the fusion bias problem in multi-view learning caused by uneven view quality. Furthermore, the design of the contrastive loss also demonstrates innovation. By jointly optimizing the consistency between independent features and fused features with the global semantics respectively, the model maintains a balance between multi-view commonality and specificity while enhancing the discriminability of the representations.

**Weaknesses:**

However, the method has several shortcomings worthy of in-depth discussion regarding its algorithm design, theoretical rigor, and technical details.
1. The L2-norm fusion strategy lacks theoretical justification and may erroneously amplify noisy views, undermining the reliability of feature integration.
2. The contrastive loss design suffers from imbalance between independent and fused terms and lacks sensitivity analysis for key parameters τ and α.
3. The experimental evaluation omits efficiency, robustness, and statistical significance analyses, limiting the credibility and generalizability of the results.

**Questions:**

1. Formula (2) assumes that the larger the feature norm, the more reliable the view. However, high-noise views may also have larger norms and thus be incorrectly weighted, amplifying the noise impact. The authors did not verify this assumption or provide a comparison with other fusion strategies, which weakens the persuasiveness of the design.
2. The unified τ and α in Formula (5) lack a basis and no sensitivity analysis is performed. More importantly, the number of independent view terms O(M) and fusion terms O(M²) is seriously unbalanced, causing the model to over-focus on fusion features and ignore view characteristics. There is no normalization or ablation verification in the paper.
3. Training efficiency, video memory consumption, or robustness under extreme noise and view loss scenarios are not reported, and there is no statistical significance analysis, making it difficult to fully demonstrate its practicality and stability.
4. In the experimental part, the author selected a total of 5 datasets, among which Fashion, Synthetic3d and Cifar10 showed little improvement compared to the second-best results, which cannot effectively demonstrate the effectiveness of the proposed method. It is recommended that the author select more convincing datasets for experimental verification.

---

### Official Review · Reviewer_9Jp3 · 2025-10-30

**Soundness:** 1
**Presentation:** 3
**Contribution:** 2
**Rating:** 4
**Confidence:** 5

**Summary:**

Existing multi-view clustering method struggle with semantic fragmentation and inefficient view integration when processing complex multi-view data. To address these challenges, this paper
proposes a Hierarchical Contrastive learning and Dynamic Attention (HCDA) fusion framework. Experimental results demonstrate HCDA’s superiority over state-of-the-art methods on  benchmark datasets. Ablation studies confirm the efficacy of proposed modules, while parameter analysis highlights robustness to hyperparameter variations.

**Strengths:**

1.The paper is well-written.
2.The authors achieve the state-of-the-art performance.
3.The experimental results are relatively comprehensive.

**Weaknesses:**

1.According to “Multi-view clustering integrates complementary features and semantics across
modalities to provide comprehensive data representations”, the goal of multi-view clustering is learning comprehensive data representations, but the output of the algorithm pseudocode is clustering indicator matrix U. So, what is the objective of multi-view clustering?
2.A critical statement regarding the authors’ motivation is: existing DMVC methods suffer from (1) insufficient modeling of fine-grained semantic interactions and hierarchical alignment and (2) ineffective multi-view integration. However, the authors provide neither empirical results nor theoretical analysis, thereby rendering the mentioned statement fragile.
3.One of the repeatedly mentioned advantages of the proposed HCDA is “robustness”, but the authors provide no experimental results (such as performance in scenarios involving noise, missing views, distribution shifts, and so on) to support this statement that HCDA is robust.
4.Indeed, I do not know the difference between the Hierarchical Cross-View Attention Weighting and vanilla attention mechanism. Using attention scores to measure contribution in multi-modal/multi-view learning is widely researched [1,2].
5.To explain Equation (2), the authors make the statement: “This L2 norm-based fusion approach ensures that views with more prominent features play a greater role in the combined representation, while still preserving the contribution of other views.” Why can L2 norm of feature vector measure the role in multi-view fusion?
6.For two views, the authors conduct multi-view fusion by the way described in Equation (2), but the authors just perform concatenation for more than two views. In scenarios involving more than two views, why do the authors not adopt the approach for multi-view fusion described in Equation (2)? According to their own statements, such a method should be more effective.
7.The authors use both M and V to denote the number of viewpoints, which is unnecessarily redundant.
8.How to set the value of $\alpha$ is not detailed in hyperparameter sensitivity analysis.
9.Individual convergence performance is meaningless. The description “the loss function value gradually decreases as training progresses and eventually converges. Clustering performance metrics (such as ACC and NMI) rise rapidly in the early stages and stabilize later” applies to the vast majority of baselines (even deep learning models).
10.The framework of the method (Figure 1) is presented in Section introduction, and Figure 1 is not cited in the main manuscript.

[1].Test-time Adaptation against Multi-modal Reliability Bias. ICLR2024.
[2].Attention Bootstrapping for Multi-Modal Test-Time Adaptation. AAAI2025.

**Questions:**

Please refer to the Weakness.

---

### Official Review · Reviewer_rDQy · 2025-10-31

**Soundness:** 2
**Presentation:** 1
**Contribution:** 2
**Rating:** 2
**Confidence:** 3

**Summary:**

This paper introduces HCDA, a hierarchical contrastive learning framework that achieves fine-grained semantic decoupling by constructing multi-level semantic spaces. The authors point out the issues of semantic fragmentation and ineffective multi-view integration in current deep multi-view contrastive learning methods. The authors further propose a dual-path attention mechanism to learn weights for cross-fused features and independent features. Extensive experiments across diverse benchmarks demonstrate its efficiency and superiority.

**Strengths:**

- The experimental setup includes multiple comparison methods and ensures fairness across different methods.
- The experiments in this paper validate the effectiveness of the proposed method.

**Weaknesses:**

- In Section 4.2, why does HCDA achieve a 10% higher accuracy on the 100leaves dataset compared to the best existing method, while the improvement on the other four datasets is within 3%? The improvement of HCDA on these datasets is not significant: 0.03% on Fashion, 0.0% on Synthetic3D, 2.2% on Caltech-5V, and 0.02% on CIFAR-10.
- The proposed method is an engineering-oriented approach that combines various existing mechanisms. The proposed contrastive loss (Equation 5) is highly similar to the contrastive loss in DealMVC (Equation 13 in reference [1]). There is a lack of clear discussion on the differences and advantages of HCDA compared with existing contrastive multi-view clustering methods.
- The experiment lacks important comparative methods, such as COMIC [2], CoMVC [3], SDMVC [4], and SiMVC [3].
- In Section 4.4, the authors do not conduct comprehensive ablation studies regarding the effectiveness of the hyperparameter alpha.
- The language expression of the paper is unclear, and there are many unnecessary paragraph breaks and excessive whitespace. For example, page 6 contains nine paragraphs, and many of them consist of only a single sentence.
- The figures in the paper are unclear. The variables X, Z, R, etc., in Figure 1 are not explicitly defined or explained. The meaning of the different colors representing categories in Figure 2 is not defined.
- Other issues: (a) The metrics clustering accuracy (ACC), normalized mutual information (NMI), and purity (PUR) lack clear explanations. (b) Several references require page numbers to meet standard citation practices.

[1] Yang, Xihong, et al. "Dealmvc: Dual contrastive calibration for multi-view clustering." Proceedings of the 31st ACM international conference on multimedia. 2023.

[2] Peng, Xi, et al. "COMIC: Multi-view clustering without parameter selection." International conference on machine learning. PMLR, 2019.

[3] Trosten, Daniel J., et al. "Reconsidering representation alignment for multi-view clustering." Proceedings of the IEEE/CVF conference on computer vision and pattern recognition. 2021.

[4] Xu, Jie, et al. "Self-supervised discriminative feature learning for deep multi-view clustering." IEEE Transactions on Knowledge and Data Engineering 35.7 (2022): 7470-7482.

**Questions:**

Please see the weaknesses.

---

### Official Review · Reviewer_jXYA · 2025-10-31

**Soundness:** 3
**Presentation:** 2
**Contribution:** 2
**Rating:** 2
**Confidence:** 4

**Summary:**

This paper proposes HCDA, a hierarchical contrastive learning framework for multi-view clustering. The method constructs three levels of semantic representations: view-specific, cross-view fused, and global. It introduces a dual-path contrastive objective to align these representations with a global semantic anchor. Additionally, an attention-based weighting mechanism dynamically adjusts the importance of each view based on its semantic relevance to the global representation. Experimental results on several benchmark datasets show competitive clustering performance compared to recent state-of-the-art methods.

**Strengths:**

**S1. Clear Problem Formulation and Motivation**

The paper clearly defines key challenges in multi-view clustering, including semantic fragmentation and ineffective view integration, and motivates the proposed structural approach as a solution to these issues.


**S2. Well-structured Architectural Design**

The architecture is clearly organized, with a hierarchical separation between view-specific, cross-view fused, and global semantic representations. The structural design follows the core principles of multi-view representation learning and supports a coherent learning flow.

**Weaknesses:**

**W1. Lack of Theoretical Justification**

While the paper introduces dual-path attention and hierarchical contrastive learning, it lacks theoretical analysis or formal justification for why these specific designs are effective for multi-view clustering. The method relies heavily on heuristic architectural choices without analyzing their impact on representation quality, alignment.


**W2. Marginal Novelty**

The paper combines hierarchical semantic representation with techniques already widely explored in prior work, such as contrastive learning with attention-based weighting. Most components, including view-specific and cross-view contrastive losses and attention-weighted fusion, are adaptations of existing methods rather than fundamentally new contributions. As a result, the overall novelty is marginal, relying more on structural integration than on genuine algorithmic innovation.


**W3. Experimental Weakness**

While the method introduces several architectural components, the empirical gains over existing baselines are limited, with notable improvement only on the Caltech-5V dataset. Performance on other benchmarks remains comparable to prior work, much of which already nearly saturated.

**W4. Methodological Validity Concern**

In Eq. (2), the authors propose an L2-norm-guided fusion to construct a fused representation. However, the two features being fused are extracted from separate view-specific encoders, which likely operate in different latent spaces with inconsistent scales. As a result, comparing their L2 norms to determine fusion weights is not theoretically justified, and may lead to a biased or semantically invalid fused representation.

**Questions:**

**Q1.**
One of the key claims is the model’s ability to suppress low-quality views via attention weighting. However, the paper does not evaluate the model under explicitly corrupted, noisy, or missing view settings. Have you tested the model’s robustness in such scenarios?


**Q2.**
You emphasize semantic disentanglement and alignment across views. Can you provide qualitative evidence, such as attention heatmaps, cluster-wise similarity plots, or latent space visualizations, to support this claim?


**Q3.**
The proposed method introduces several additional components, including pairwise view fusion, dual-path attention, and contrastive objectives. Have you analyzed the impact of these components on computational cost compared to existing methods? Can the method scale efficiently with increasing view numbers or larger datasets?

---

### Official Review · Reviewer_KLrD · 2025-10-31

**Soundness:** 3
**Presentation:** 2
**Contribution:** 2
**Rating:** 4
**Confidence:** 4

**Summary:**

This paper addresses semantic fragmentation and inefficient view integration in multi-view data by proposing a Hierarchical Contrastive learning and Dynamic Attention (HCDA) framework. HCDA constructs multi-level semantic spaces for fine-grained semantic decoupling and employs a dual-path attention mechanism to adaptively weight views based on global semantics. By jointly optimizing contrastive losses between cross-view and independent features, it balances view commonality and specificity. Experiments and ablation studies validate the effectiveness of the proposed framework.

**Strengths:**

1. The dual-path attention mechanism adaptively emphasizes high-quality views and suppresses noisy ones.

2. Representative baseline methods are employed for comparison, and the results demonstrate the effectiveness of the proposed approach.

**Weaknesses:**

1. Given the abundance of multi-view clustering literature employing contrastive learning and attention-based methods, the novelty of this paper appears to be limited.

2. The experimental data lacks a tabular summary.

3. In the experimental section, apart from Table 1 which covers all datasets, the remaining experiments were conducted only on selected datasets, thereby lacking sufficient persuasiveness.

4. The tables are placed after the conclusion, which violates standard formatting conventions.

**Questions:**

See Weaknesses

---

### Note · Authors · 2025-12-04

I have read and agree with the venue's withdrawal policy on behalf of myself and my co-authors.